# The Effect of Application of Chicken Gelatin on Reducing the Weight Loss of Beef Sirloin after Thawing

**DOI:** 10.3390/polym14153094

**Published:** 2022-07-29

**Authors:** Jakub Martinek, Robert Gál, Pavel Mokrejs, Kristýna Sucháčková, Jana Pavlačkova, Alena Kalendová

**Affiliations:** 1Department of Polymer Engineering, Faculty of Technology, Tomas Bata University in Zlín, Vavrečkova 5669, 760 01 Zlín, Czech Republic; mokrejs@utb.cz (P.M.); kalendova@utb.cz (A.K.); 2Department of Food Technology, Faculty of Technology, Tomas Bata University in Zlín, Vavrečkova 5669, 760 01 Zlín, Czech Republic; gal@utb.cz (R.G.); k_suchackova@utb.cz (K.S.); 3Department of Lipids, Detergents and Cosmetics Technology, Faculty of Technology, Tomas Bata University in Zlín, Vavrečkova 5669, 760 01 Zlín, Czech Republic; pavlackova@utb.cz

**Keywords:** freeze–thaw loss, biobased packaging, coating, poultry gelatin, beef sirloin, freezing, texture, color, pH, meat quality

## Abstract

Freezing is one of the oldest and most-often-used traditional methods to prolong the shelf life of meat. However, the negative phenomenon of this process is the weight loss of water that occurs after the meat is thawed. Together with the water that escapes from the meat during thawing, there are large weight losses in this valuable raw material. Another negative aspect is that mineral and extractive substances, vitamins, etc. also leave the meat, resulting in irreversible nutritional losses of nutrients in the meat, which are subsequently missing for use by the consumer of the meat. The main goal of this work is to reduce these losses by using gelatin coatings. Gelatin was prepared from chicken paws according to a patented biotechnological procedure, which uses the very gentle principle of obtaining gelatin with the usage of enzymes. This unique method is friendly to the environment and innocuous for the product itself. At the same time, it ensures that the required principles achieve a circular economy with the use of the so far very-little-used slaughter byproducts, which in most parts of the world end up in uneconomic disposal by burning or landfilling without using this unique potential source of nutrients. Gelatin coatings on the surface of the beef steak were created by immersing the meat in a solution based on gelatin of different composition. A coating containing 3%, 5% or 8% gelatin with 10% or 20% glycerol (by weight of gelatin) and 1% glutaraldehyde crosslinker (by weight of gelatin) has proved to be effective. The amount of glutaraldehyde added to the coating is guaranteed not to exceed the permitted EU/U.S. legislative limits. In addition to weight loss, meat pH, color and texture were also measured. Freezing was done in two ways; some samples were frozen at a normal freezing temperature of −18 °C and the other part of the experiment at deep (shock) freezing at −80 °C. Defrosting took place in two ways, in the refrigerator and in the microwave oven, in order to use the common defrosting methods used in gastronomy. A positive effect of this coating on weight loss was observed for each group of samples. The most pronounced effect of coating was found for the least gentle method of freezing (−18 °C) and thawing (microwave), with the average weight loss of the coated samples differing by more than 2% from that of the uncoated sample. No negative effect of the coating was observed for other meat properties tested, such as pH, Warner-Bratzler Shear Force (WBSF) or color. Gelatin-based coating has a positive effect on reducing the weight loss of meat after thawing. Chicken gelatin prepared by a biotechnological process has a new application in improving the quality of meat due to the retention of water and nutrients in frozen and subsequently thawed beef, which can contribute to the better quality of the subsequently gastronomically prepared dish, while maintaining the weight and nutritional quality. This also results in economic savings in the preparation of highly-valued parts of beef.

## 1. Introduction

Freezing meat is one of the possible technological steps that can be used to extend its shelf life, due to the inhibition of unwanted microorganisms that are responsible for meat spoilage [1]. Unfortunately, this process is accompanied by side effects such as the formation of ice crystals in the structure of the meat. After thawing, such a damaged structure cannot keep the meat juice inside, and water, proteins, minerals, vitamins, and other important extractive substances are lost [2,3].

The rate of freezing affects the formation of ice crystals and consequently the quality of the thawed meat. It is important to freeze the meat as quickly as possible, both to increase shelf life and to reduce weight loss after thawing. In general, the lower the freezing temperature and the higher the freezing rate, the lower the weight loss after thawing. During very fast freezing, small ice crystals of approximately the same size are formed, and due to the presence of small ice crystals, the muscle structure is not significantly disturbed. This is associated with the risk of protein denaturation and lipid oxidation, as a result of which meat ceases to be edible [4]. At higher temperatures, on the other hand, large crystals of different sizes are formed, which disrupt the structure of the meat, which results in weight loss after thawing. Commercially available freezers operate at −18 °C, which is considered a slower method of freezing due to the freezing rate. For correct and efficient freezing, it is necessary to consider, among other things, the temperature of the cooling medium, the weight of the carcasses, the fat cover, which serves as thermal insulation, and the size of the gaps between the individual pieces of meat [5].

Food coatings are used to protect food until it is consumed by the consumer. In the case of coatings, great emphasis is placed on protection against physical influences, mechanical damage, and high-quality barrier properties, where it is primarily about the degree of permeability to water vapor, O_2_ and CO_2_ [6]. In addition, edible coatings have the advantage of being edible and easily biodegradable [7]. The main raw material for the formation of coatings are polymers, edible proteins, and polysaccharides. Proteins can be used of both animal origin (collagen, gelatin, keratin, egg, and milk proteins) and vegetable origin (soy protein, cereal gluten, corn zein) [8]. The barrier properties of protein-based coatings depend on the amounts and arrangement of amino acids in the protein. Of the polysaccharides, those whose monosaccharide units are linked by β-(1,4) glycosidic bonds can be used to form coatings. Examples include cellulose and its modifications (hydroxypropylcellulose, methylcellulose, and carboxymethylcellulose), chitosan, and starch [9]. To improve the mechanical properties, it is necessary to add other substances, such as plasticizers or crosslinkers. Plasticizers (saccharides, polyols, etc.) reduce the brittleness of the coating, which makes it more flexible. Crosslinkers (such as aldehydes, enzymes, UV, etc.), in turn, increase the strength and durability of the coating by creating a more stable spatial network [10]. Glutaraldehyde is widely used solely to crosslink gelatin. Nevertheless, it may have cytoxic effects at higher concentrations. Therefore, maximum acceptable limit of 2 mg glutaraldehyde per one kilogram of product is allowed by EU legislation [11]. US legislation allows the use of glutaraldehyde as a crosslinker for food contact materials [12]. In addition to crosslinkers and plasticizers, it is possible to incorporate other active substances (e.g., antioxidants or substances with antimicrobial properties) into the coating, which bring other desired properties [13], e.g., essential oils (tea tree, rosemary, clove, lemon, oreganum, etc.) and bioactive compounds (bee pollen, ethanolic extract of propolis, pomegranate dried extract etc.) [14], phenolic acids (gallic acid, p-hydroxy benzoic acid or ferulic acids) or flavonoids (catechin, flavone or quercetin) [15].

The most common methods of coating are panning, fluidized bed, spraying and dipping. Panning takes place in a large rotating drum; the coating solution is sprayed into the rotating drum with the object to be coated, and thanks to the rotation of the drum, the application of the coating is uniform. When using a fluidized bed, the preheated object is placed in a space with dispersed coating particles. Spray application is a more controllable method than using a fluidized bed. Spraying ensures even coating of the product surface. Dipping consists of immersing the food in the coating solution. This method is suitable for irregularly shaped objects [16,17].

In practice, various types of edible packaging are often used. After thawing, they can help keep the juice in the meat and thus reduce water losses during thawing or even heat treatment of the meat. The literature describes the application of edible polymer coatings based on gelatin, or composite coatings with the addition of other polymers (e.g., cellulose, methylcellulose, etc.). However, pork or beef gelatin is used in all studies. The traditional raw materials to produce gelatin are of beef and pork origin (skins, bones, etc.). However, new, alternative sources of collagen from fish and poultry are also beginning to be used. Even though methods for preparation from alternative raw materials (fish, poultry) are known, the application of these gelatins to edible polymer coatings is not yet mentioned [18]. The advantage of edible or biodegradable coatings is their environmental friendliness, as they reduce the consumption of plastics. In addition, the gelatin prepared by us uses byproducts from poultry processing as a source, which the industry often cannot deal with other than by incineration [19].

Other studies mention that by adding other substances to the coating of natural origin, the addition of artificially prepared substances to the food itself can be eliminated.

These can be natural antioxidants obtained from plants (e.g., Caesalpinia decapetala or Caesalpinia spinosa Tara), which will extend shelf life [20]. Further research, in addition to prolonging shelf life, was also interested in the possibility of improving taste by coating based on onion extract [21]. The effect of propolis added to the gelatin coating on shelf life was also observed. In addition to the antioxidant properties of the coating, the inhibitory effects on the growth of microorganisms were also observed [22].

The novelty of this research is based on the material that is used to produce the coating. It is gelatin that was obtained from the by-products of poultry processing, which is an unconventional and less-used source of collagen in practice, by gentle enzymatic hydrolysis of collagen. Not only do we use byproducts that have no other significant use as a source of gelatin, but also the gelatin is prepared under environmentally gentle biotechnological process not using acids or alkalis as common in industrial gelatin manufacturing practice [23].

The aim of this paper is to reduce the weight loss of beef after thawing by applying a polymer solution based on gelatin prepared from chicken byproducts according to a previously published technological procedure [23] and to apply the polymer solution to beef low sirloin steaks by dipping. The raw material to produce the edible coating used in this paper is gelatin obtained by a bio-technological process from collagen contained in chicken paws. The task in the coated meat samples was to monitor weight losses and other selected qualitative parameters depending on the different methods of freezing and thawing. The scientific hypothesis is that the application of a gelatin-based polymer coating and the choice of a suitable freezing and thawing method will reduce weight loss.

Thanks to this, sensory active substances, vitamins, minerals, and proteins inside the meat would be preserved. Another significant benefit in confirming the hypothesis would be the possibility of using lower freezing temperatures, which would otherwise cause greater weight loss due to the formation of larger ice crystals. This effect would then have a positive economic impact due to lower energy consumption.

## 2. Materials and Methods

### 2.1. Raw Materials and Chemicals

Beef low sirloin (Steinhauser Ltd., Tišnov, Czech Republic) gelatin prepared by a biotechnological process from chicken paws according to Patent CZ 307665—Biotechnology-based production of food gelatin from poultry byproducts, glycerol (Fichema, Brno, Czech Republic) and glutaraldehyde (Penta Ltd., Praha, Czech Republic).

### 2.2. Appliance and Tools

Scale Kern 440-49 N (Kern & Sohn GmbH, Balingen, Germany), vacuum packaging machine Mini Jumbo (Henkelman, Hertogenbosch, The Netherlands), textural device TA-XT Plus (Stable Micro Systems, Surrey, UK), calorimetr Ultrascan PRO (HunterLab, Reston, VA, USA), heating convector Racional SCC WE 61 (RATIONAL, Praha, Czech Republic), pH metr Hanna HI 99161 (Hanna Instruments Czech Ltd., Praha, Czech Republic), refrigerator (Liebherr, Bulle, Switzerland), freezer (Arctiko A/S, Esbjerg, Denmark), microwave (Whirlpool, Benton Charter Township, MI, USA), bags for vacuum packaging, other laboratory and kitchen equipment.

### 2.3. Processing of Chicken Paws into Gelatin

Separation of soluble proteins (albumins, globulins) from chicken paws was performed according to the procedure from a previously published work, only with minor modifications [24]. The tissue was then defatted in a 1:1 mixture of ethanol and petroleum ether. The raw material was shaken with the above mixture in a ratio of 1:6 for three days, with the solvent being changed every 24 h.

Purified collagen was mixed with distilled water in a ratio of 1:10; the pH was adjusted to 6.5–7.0. Then the proteolytic enzyme Protamex was added in an amount of 0.4% to the collagen dry matter, and the mixture was shaken for 15 h. Conditioned collagen was mixed with distilled water in a ratio of 1:8; gelatin was extracted at 65 °C for 4 h; gelatin solution was dried at 55 °C in a thin film [23].

### 2.4. Coating Preparation

The coating was prepared by dissolving 3%, 5%, 8% (*w/v*) gelatin in water with constant heating (70 ± 1.0 °C for 20 min) and stirring. Subsequently, glycerol was added to the solution at a concentration of 10% and 20% by the amount of gelatin. Glycerol was added during cooking after the gelatin had dissolved. One coating formulation also contained glutaraldehyde at a concentration of 1% by weight of gelatin, which was added before the end of cooking. The total amount of glutaraldehyde was determined so as not to exceed the permitted food contact limits [11,12].

### 2.5. Sample Preparation

Steaks approximately 2.5 cm thick were prepared from a matured (2 months) beef low sirloin from a 19-month-old bull. After slicing the steaks, the pH of the chilled meat was measured using a puncture pH meter, and the color and texture of the meat were also observed. Subsequently, a gelatin coating was formed on the steak by soaking in a gelatin solution for 10 s and solidifying at 4 °C for 15 min.

The meat samples thus prepared were weighed and packed using a vacuum device to partially extract air (−0.6 bar). The packaged samples were frozen at two freezing temperatures (−18.0 ± 2.0 °C, −80.0 ± 2.0 °C) and stored at this temperature for 14 days. A portion of the samples was thawed in a refrigerator at the temperature of 4.0 ± 2.0 °C, and other samples were thawed one-by-one in a microwave oven, where the thawing process with a power of 100W took approximately 4 min (the calculation of the required time is performed by the device automatically depending on the weight of the meat). A part of the samples was heat-treated in a convection oven, when the steak core reached the temperature of 70.0 ± 0.5 °C; this temperature was applied for 10 min. The reason for the heat treatment was to compare the weight losses and verify the protective effects of the coatings.

Another part of the thawed meat samples was further used to measure parameters that could be affected by the low freezing temperature. 

### 2.6. pH Measuring

The pH was first measured for meat chilled without a protective gelatin coating. After thawing the test samples, the pH of each sample was measured. Each sample was measured six times in different sections and at different depths to consider the volume of the whole steak as much as possible. 

### 2.7. Textural Properties Measuring

The textural properties were measured with a Warner-Bratzler knife. This method is a simulation of the first bite of food in the mouth. It is a measurement of the shear force (WBSF), i.e., the force required to cut the sample. Sliced meat samples measuring 2.0 × 2.0 × 1.5 cm were placed on the table of the TA-XT Plus Texturometer so that the knife was able to cut the sample perpendicularly to the direction of the fibers. The knife speed was set at 1 mm/s. The device records the force that must be applied to divide the individual meat samples [25].

### 2.8. Color Measuring

The color of the chilled and thawed meat was measured in three-dimensional space CIE L*a*b*. The coordinates L*a*b* describe color as a point in three-dimensional space. L* represents the brightness of the color and takes values from 0 (black) to 100 (white), a* determines the color from green (−a*) to red (+a*), the last measured parameter b* determines the color range from blue (−b*) to yellow (b*). The parameters a* and b* are located in the horizontal plane and L* in the vertical plane. Samples were analyzed on an Ultrascan PRO (HunterLab, Reston, VA, USA) with a d65 light source and a 0.19-inch diameter. Individual samples were measured six times at different locations.

### 2.9. Weight Loss Measuring

Prepared steaks were individually weighed to two decimal places. Samples were reweighed after thawing. Weight losses were calculated from the weight of the meat before freezing and after thawing according to the Equation (1):
$$WL=\frac{m_{F}-m_{D}}{m_{F}}\cdot 100$$

*WL* is the weight loss (%), *m_F_* is weight before freezing, *m_D_* is weight after defrosting.

### 2.10. Statistical Analysis

For each sample, pH, color and texture were measured six times. The weight loss was calculated according to the Formula (1) given in Section 2.8. All standard deviations, calculations, tables and graphs were performed using Microsoft Office Excel 365 (Microsoft, Denver, CO, USA). Statistical analysis was performed using XLSTAT (Addinsoft, New York, NY, USA) software using multivariate analysis of variance (MANOVA) (α = 0.05).

## 3. Results and Discussion

Table 1 and Table 2 show the results of the statistical analysis, which are then presented for the results of the tested properties.

In the rows of the remaining Table 3, Table 4, Table 5, Table 6, Table 7, Table 8, Table 9 and Table 10, there are individual combinations of freezing temperatures and thawing in the refrigerator (R) or microwave oven (MW). The values in the columns are arranged according to the composition of the coating used so that the numbers separated by slashes represent the gelatin/glycerol (alternatively gelatin/glycerol/glutaraldehyde) content.

Coating with crosslinker had the same composition in all cases, with four coated and one uncoated sample (0/0/0) for each combination of freezing and thawing. The tables for pH, texture and color do not give an average value for the coated samples to show that the measurement results are not dependent on the presence of the coating.

### 3.1. Statistical Analysis

Table 1 and Table 2 contain tests performed in multivariate analysis of variance (MANOVA). The results of the analysis are used in the comments on the results of individual monitored parameters.

**Table 1 polymers-14-03094-t001:** MANOVA results of the coatings without crosslinker.

	Wilks’ Test	Hotelling–Lawley’s Test	Pillai’s Test	Roy’s Test
Lambda	0.150	2.756	1.386	1.341
F Observed values	1.089	1.047	1.149	2.904
DF1	24	24	24	6
DF2	36	16	52	13
F Critical value	1.823	2.220	1.729	2.915
*p*-value	0.401	0.472	0.330	0.051

**Table 2 polymers-14-03094-t002:** MANOVA results of the coatings with crosslinker.

	Wilks’ Test	Hotelling–Lawley’s Test	Pillai’s Test	Roy’s Test
Lambda	0.619	0.616	0.381	0.616
F Observed values	1.335	1.335	1.335	1.335
DF1	6	6	6	6
DF2	13	13	13	13
F Critical value	2.915	2.915	2.915	2.915
*p*-value	0.310	0.310	0.310	0.310

As the computed *p*-value is greater than the significance level α = 0.05, one cannot reject the null hypothesis that the coating has no significant effect on the values obtained by measurements.

### 3.2. pH

The pH values of the low sirloin steaks are given in Table 3 and are divided according to the type of coating used, the freezing temperature and the thawing method. The pH, which was measured before the start of the experiment, was 5.12 ± 0.05.

**Table 3 polymers-14-03094-t003:** pH ± SD values of coated and uncoated meat samples depending on the method of freezing and thawing.

Freezing/Thawing	Gelatin/Glycerol (%)				
	3/10	3/20	5/10	5/20	0/0
−18 °C/R	4.70 ± 0.05	4.65 ± 0.22	4.97 ± 0.05	5.09 ± 0.02	5.08 ± 0.05
−80 °C/R	5.20 ± 0.08	5.16 ± 0.02	5.16 ± 0.03	5.11 ± 0.01	5.16 ± 0.05
−18 °C/MW	5.12 ± 0.09	5.16 ± 0.08	5.14 ± 0.03	5.20 ± 0.07	5.20 ± 0.04
−80 °C/MW	5.14 ± 0.02	5.27 ± 0.02	5.16 ± 0.03	5.18 ± 0.06	5.23 ± 0.04

R—refrigerator, MW—microwave oven.

Among all samples, in those that were frozen at −18 °C and thawed in the refrigerator, the combination of 3% gelatin coating was proved to have greatest effect on the pH. At freezing temperature of −18 °C and thawing in the refrigerator, the lowest pH was 4.65 ± 0.22 for the sample, on which a protective coating with 3% concentration of gelatin and 20% glycerol was applied. In contrast, the highest pH was 5.09 ± 0.22 with a sample of 5% gelatin and 20% glycerol.

Steaks that were frozen at −80 °C and thawed in the refrigerator showed the lowest pH of 5.11 ± 0.01. In this case, it was a sample with a coating consisting of solution of 5% gelatin concentration with 20% glycerol. The highest value was 5.20 ± 0.08, where the protective coating consisted of 3% gelatin together with 10% glycerol.

The pH values of meat frozen at −18 °C and thawed in a microwave oven differed from the values of meat frozen at the same temperature and thawed in a refrigerator. The lowest pH was measured on a sample coated with 3% gelatin and 10% glycerol. The highest pH was for the steak with a protective coating consisting of 5% gelatin and 20% glycerol. The same pH was measured on a sample to which no gelatin coating was applied.

The pH results from the last freezing and thawing method (−80 °C, MW) are given in the last row of Table 3. Here, the lowest pH 5.14 ± 0.02 was found on a sample with a protective coating of the same composition as on a sample frozen at −18 °C and thawed in the microwave. The specific composition of this coating was 3% gelatin concentration and 10% glycerol. In contrast, the highest pH was in the sample with 3% gelatin concentration and 20% glycerol.

The application of the coating generally did not have effect on pH, and the dependence of the amount of gelatin and glycerol on the pH cannot be deduced from the values obtained. Only three samples gave pH values lower than 5.00, the others were around the pH of the meat, which was measured before the start of the experiment.

Table 4 shows the pH values of the coating with crosslinker and uncoated samples and is divided according to freezing temperature and thawing method.

**Table 4 polymers-14-03094-t004:** The pH ± SD values of the meat samples with crosslinked coating and without coating depend on the method of freezing and thawing.

Freezing/Thawing	Gelatin/Glycerol/Glutaraldehyde (%)				
	8/10/1				0/0/0
−18 °C/R	4.90 ± 0.08	4.90 ± 0.06	5.03 ± 0.04	5.22 ± 0.32	5.17 ± 0.06
−80 °C/R	5.07 ± 0.60	5.39 ± 0.06	5.38 ± 0.10	5.14 ± 0.03	5.06 ± 0.08
−18 °C/MW	5.17 ± 0.09	5.16 ± 0.08	5.17 ± 0.07	5.19 ± 0.07	5.06 ± 0.04
−80 °C/MW	5.19 ± 0.07	5.17 ± 0.02	5.25 ± 0.03	5.14 ± 0.03	5.19 ± 0.03

R—refrigerator, MW—microwave.

The pH of the meat thawed in the refrigerator ranged from 4.90 ± 0.08–5.39 ± 0.06. The pH values of the samples that were thawed in the microwave were like the previous values. The lowest pH value in this group of samples was set at 5.06 ± 0.08 and the highest 5.25 ± 0.03. However, the pH of the meat in the other samples is not different from the pH measured from the first part of the experiment, in which other gelatin coatings were used.

Jaberi et al. investigated the effect of vacuum and modified atmospheres on the technological properties of meat. The pH values of the meat ranged between 5.65 and 5.75. The measured values of our samples are slightly more acidic, but the difference is not significant, and the values correspond to used kind of meat cuts [26]. Cheng et al. investigated how repeated freezing and thawing affects meat quality in terms of water quantity and distribution. One of the observed characteristics was the pH whose value after the first thawing was around 5.73. This is a higher value than in our experiment, but the same trend is important, which showed that after the first thawing, the pH value did not change significantly from that found before freezing [25]. Sales et al. compared technological parameters such as weight loss after thawing, pH, WBSF, represented by shear force, and color in chilled, thawed, and radiation-treated beef meat. Even in this case, the pH values were generally higher than in this study, but even here the pH of the meat before freezing and after thawing did not differ significantly [27]. Rahman et al. investigated the effect of repeated freezing and thawing on the chemical–physical parameters of back beef. Before freezing, the meat had a pH of 6.15, and after the first cycle of freezing and thawing, its amount did not change significantly (a decrease of 0.2) [28]. Kim et al. evaluated the effect of aging and freezing and thawing cycles on color and other physicochemical and enzymatic properties of two bovine muscles (*Mm. gluteus medius a biceps femoris*). Values of pH were among the tested parameters. However, even in this case, no significant deviation in its value was confirmed for meat before and after freezing [29]. Aroeira et al. tested the influence on tenderness of meat freezing of young Nellore and Aberdeen Angus bull meat before ageing. The monitored parameters included pH; however, pH changes when comparing pre-freeze and post-thaw values were minimal (a decrease of 0.03) [30].

The literature mentions various pH ranges; however, from the microbiological safety point of view, the pH should not be lower than 4.5. The pH suitable for technological processing of meat should be in the range between 5.2 and 5.7 [25,27,28]. All measured values comply with meat safety demands and are acceptable for further technological processing.

### 3.3. Texture

The WBSF of the meat was determined from the textural properties. The values obtained are given in Table 5 and are divided according to the type of coating used, the freezing temperature and the thawing method. The shear force before the start of the experiment was 33.05 ± 7.77 N.

**Table 5 polymers-14-03094-t005:** Meat WBSF values ± SD [N] with and without coating depending on the method of freezing and thawing.

Freezing/Thawing	Gelatin/Glycerol (%)				
	3/10	3/20	5/10	5/20	0/0
−18 °C/R	32.90 ± 6.49	31.71 ± 4.71	46.57 ± 2.44	35.93 ± 9.26	15.14 ± 4.82
−80 °C/R	35.91 ± 10.38	35.91 ± 8.24	35.53 ± 11.48	34.74 ± 9.04	45.59 ± 16.42
−18 °C/MW	37.28 ± 12.52	33.48 ± 7.70	36.99 ± 20.88	38.48 ± 7.16	31.87 ± 12.67
−80 °C/MW	34.75 ± 23.23	40.46 ± 20.68	51.12 ± 13.20	46.73 ± 15.35	40.03 ± 10.40

R—refrigerator, MW—microwave oven.

The general assumption is that after thawing, the shear force of the meat should be lower than before the start of the experiment, because freezing disrupts the structure of the meat. The meat should therefore be more tender after thawing. The measured values did not confirm this assumption.

For samples frozen at −18 °C and thawed in the refrigerator, the lowest shear force was measured for the sample without the gelatin coating. On the contrary, the highest shear force was again measured for steak with a protective coating which contained 5% gelatin and 10% glycerol.

The opposite effect can in turn be observed for shear force values obtained from samples frozen at −80 °C and thawed in the refrigerator. The lowest shear force was determined for a 5% gelatin coating with 20% glycerol. The highest value, which exceeded 45.586 ± 16.421 N, was measured on the uncoated sample.

The meat frozen at −18 °C, which was thawed in a microwave oven, showed approximately the same shear force values as in the previous case. The lowest value was again, as in the first case in the blank sample, i.e., in the steak, which was not coated with a gelatin coating. The highest WBSF was recorded for a coated sample with 5% gelatin and 20% glycerol.

The lowest shear force value of the samples, which were frozen at −80 °C and thawed in a microwave oven, was measured in meat with a 3% gelatin coating, in which there was less glycerol, namely 10%. The value that exceeded 51 N was the highest value not only in this method of freezing and thawing, but it was also the highest of all measurements. The values could be influenced by the nature of the meat, which means that the meat could contain more tenders in certain parts, and therefore, the measured sample could be harder than the other samples. From the obtained measurements, it cannot be unequivocally stated that the gelatin coatings influenced the shear force of the meat.

Table 6 shows the WBSF values for the coated (with addition of crosslinkers) and uncoated samples (0/0/0) and is divided according to the freezing temperature and the thawing method.

**Table 6 polymers-14-03094-t006:** Meat WBSF values ± SD [N] with crosslinked coating and without coating depending on the method of freezing and thawing.

Freezing/Thawing	Gelatin/Glycerol/Glutaraldehyde (%)				
	8/10/1				0/0/0
−18 °C/R	21.50 ± 7.11	22.69 ± 8.41	23.61 ± 1.47	33.76 ± 11.05	24.98 ± 4.36
−80 °C/R	26.32 ± 7.61	20.77 ± 6.46	29.48 ± 12.79	25.98 ± 4.03	30.87 ± 8.34
−18 °C/MW	22.41 ± 3.46	23.40 ± 8.29	47.45 ± 20.28	25.10 ± 5.79	29.36 ± 4.48
−80 °C/MW	26.68 ± 11.01	37.29 ± 14.11	23.77 ± 5.60	28.69 ± 8.86	27.21 ± 7.39

R—refrigerator, MW—microwave oven.

For this set, the values found were generally lower than in the previous table and also lower compared to the WBSF before the start of the experiment. There is no apparent dependence of shear force on the use of the coating or the method of freezing and thawing of the samples. The values obtained ranged from 20.77 ± 6.46 N to 47.45 ± 20.28 N, with both extreme values being found in the coated sample. It depended on the structure of the test sample rather than the coating under investigation.

Cama-Moncunill et al. indicate that the value of meat WBSF measured with a Warner-Bratzler knife is optimal in the range of 40–45 N. Lower values are typical for meat that is fine, tender and properly matured. In the experiment, the WBSF values obtained by measuring with a Warner-Bratzler knife ranged from 30 to 50 N [31]. Cheng et al., in contrast to our study, observed a trend of increasing meat shear force after the first thawing; however, the increase could be due to a different type of meat [25]. Sales et al. found in chilled meat that the value of shear force is almost double in comparison with the highest result in our study, while after thawing the shear force decreased slightly [27]. Kim et al. concluded in their study that meat WBSF should decrease by disrupting the structure of muscle fibers. In addition to the freezing and thawing effects, this study also compared the effect of meat maturation [29]. In Aroeira et al., the values before freezing were around 60 N, and after thawing, the shear force decreased only slightly. These shear force values were different compared to our study. The reason for this difference may be the fact that they used immature meat, which is generally firmer [32].

Based on the obtained values, no negative effect of the coating on the textural properties of the meat was observed.

### 3.4. Color

The obtained color values according to the space defined by the brightness and the color axes a* and b* are given in Table 7 and sorted according to the type of coating used, the freezing temperature and the thawing method. The pre-freeze color values for brightness were 39.36 ± 0.79, half-axis a* 12.88 ± 0.89 and half-axis b* 7.09 ± 0.62.

**Table 7 polymers-14-03094-t007:** Coated and uncoated meat color values ± SD depending on the method of freezing and thawing.

Freezing/Thawing		Gelatin/Glycerol (%)				
		3/10	3/20	5/10	5/20	0/0
−18 °C/R	L*	25.97 ± 0.63	21.16 ± 0.50	23.13 ± 2.25	26.99 ± 2.18	23.91 ± 1.05
	a*	17.20 ± 1.72	20.99 ± 1.02	18.31 ± 2.67	16.41 ± 2.06	20.36 ± 1.86
	b*	18.61 ± 1.10	18.81 ± 1.41	18.19 ± 0.45	20.83 ± 2.16	20.72 ± 0.99
−80 °C/R	L*	25.97 ± 0.84	20.25 ± 1.03	24.57 ± 0.90	22.56 ± 0.83	23.21 ± 1.14
	a*	16.51 ± 1.56	20.25 ± 1.24	18.03 ± 1.57	22.76 ± 1.73	23.45 ± 1.19
	b*	16.72 ± 1.22	21.73 ± 0.81	20.11 ± 1.24	23.08 ± 0.51	22.52 ± 0.72
−18 °C/MW	L*	29.02 ± 0.87	25.87 ± 1.18	21.31 ± 0.69	24.35 ± 1.85	23.61 ± 2.12
	a*	17.47 ± 0.48	18.52 ± 1.21	25.63 ± 1.45	22.00 ± 4.36	19.95 ± 1.79
	b*	19.38 ± 0.46	18.47 ± 1.52	24.51 ± 0.89	21.93 ± 3.59	19.14 ± 1.32
−80 °C/MW	L*	27.75 ± 1.67	23.39 ± 1.23	24.31 ± 1.40	22.00 ± 2.41	24.24 ± 0.78
	a*	16.91 ± 1.58	21.84 ± 2.48	17.45 ± 2.88	20.61 ± 3.24	19.83 ± 1.47
	b*	18.27 ± 1.05	20.29 ± 3.29	18.85 ± 1.75	20.93 ± 2.16	20.03 ± 1.16

R—refrigerator, MW—microwave oven.

The lowest brightness of 21.16 ± 0.50 was for the meat samples frozen at −18 °C and thawed in the refrigerator. This steak was coated with a protective coating of 3% gelatin and 20% glycerol. The highest brightness of this group was measured in a sample with a 3% gelatin coating, but this time it contained a lower glycerol content. The lowest a* value was for steak coated with a solution of 5% gelatin and 20% glycerol. The most pronounced red color was measured in a meat sample with 3% gelatin concentration and 10% glycerol. The lowest value of b* (yellow-blue shades) was measured in the first set of samples for steak with a 5% gelatin coating, the composition of which contained 10% glycerol. The highest value was also determined for a 5% gelatin coating, except that it contained 20% glycerol.

Even the samples, which were frozen at −80 °C and thawed at 4 °C, do not show the effects of the coatings. The lowest brightness value was measured for steak, which was soaked in a 5% gelatin coating with 10% glycerol before freezing. The highest brightness value was determined for the same sample as in the previous two cases; it was the value that was the highest of all measurements performed. The lowest a* value was observed for a sample to which a coating of 3% gelatin and 10% glycerol was applied before freezing. The highest a* was measured for the sample without the presence of a protective coating. The b * values differed more from the previous freezing method. The lowest value was again measured for a sample with 3% gelatin and 10% glycerol. The highest value was again affected by a coating of 5% gelatin and 20% glycerol.

In the case of meat frozen at −18 °C and thawed in a microwave oven, the brightness was still in the range of 20–26. The lowest value was again measured for a sample coated with 3% gelatin and 20% glycerol. The highest value was again affected by a coating of 3% gelatin and 10% glycerol. The lowest a* value was observed for the sample, which also contained a protective coating, namely 3% gelatin and 10% glycerol. The most pronounced red color was found in the coated sample, which was prepared by mixing 5% gelatin with 10% glycerol. The lowest b * value was found for steak, which was soaked in a 3% gelatin coating with 20% glycerol before freezing. The biggest b* value was determined for a sample with 5% gelatin and 10% glycerol.

In the group of samples frozen at −80 °C and thawed in a microwave oven, the lowest value occurred in the case of a sample with 5% gelatin and 20% glycerol. The highest value of L* occurred in the same case. The intensity of the red color was least pronounced in the sample with 3% gelatin coating and 10% glycerol. In contrast, the most pronounced red color was in meat with 3% gelatin and 20% glycerol. On the yellow-blue axis, the values for meat fell in the yellow area. The least pronounced yellow coloration was noted in the sample with 3% gelatin coating and 10% glycerol. The most pronounced yellow coloration was found in meat with 5% gelatin and 20% glycerol.

Table 8 shows the color values for the samples with cross-linked coating and uncoated samples (0/0/0) and is divided according to the freezing temperature and the thawing method.

**Table 8 polymers-14-03094-t008:** Color values ± SD of meat with and without cross-linked coating depending on the method of freezing and thawing.

Freezing/Thawing		Gelatin/Glycerol/Glutaraldehyde (%)				
		8/10/1				0/0/0
−18 °C/R	L*	25.97 ± 0.63	29.02 ± 0.87	21.16 ± 0.49	25.87 ± 1.18	23.13 ± 2.25
	a*	17.20 ± 1.72	17.47 ± 0.48	20.99 ± 1.02	18.52 ± 1.21	18.31 ± 2.67
	b*	18.61 ± 1.10	19.38 ± 0.46	18.81 ± 1.41	18.47 ± 1.52	18.19 ± 0.46
−80 °C/R	L*	25.97 ± 0.84	27.75 ± 1.67	20.25 ± 1.04	23.39 ± 1.24	24.57 ± 0.90
	a*	16.51 ± 1.56	20.25 ± 1.24	18.03 ± 1.57	22.76 ± 1.73	23.45 ± 1.19
	b*	16.72 ± 1.22	18.27 ± 1.05	21.73 ± 0.81	20.29 ± 3.29	22.52 ± 1.24
−18 °C/MW	L*	21.31 ± 0.69	26.99 ± 2.18	24.35 ± 1.85	23.91 ± 1.05	23.61 ± 2.12
	a*	25.63 ± 1.45	16.41 ± 2.06	22.00 ± 4.36	22.35 ± 1.86	19.95 ± 1.79
	b*	24.50 ± 0.89	20.86 ± 2.16	21.93 ± 3.59	20.72 ± 0.99	19.14 ± 1.32
−80 °C/MW	L*	24.31 ± 1.40	22.56 ± 0.83	22.00 ± 2.41	23.21 ± 1.14	24.24 ± 0.78
	a*	17.45 ± 2.88	22.76 ± 1.73	20.61 ± 3.24	23.45 ± 1.19	19.8 ± 1.47
	b*	18.85 ± 1.75	23.08 ± 0.51	20.93 ± 0.76	22.52 ± 0.76	20.03 ± 1.16

R—refrigerator, MW—microwave oven.

Even in the second set, which compared samples with and without crosslinked coating, the effect of the coating on the value of all three parameters describing the color was not apparent. The brightness decreased in all samples, regardless of whether the coatings were applied or not. The remaining parameters increased their value and thus moved more into the red (for a*) and yellow (for b*) spectra compared to the previous Table 5.

According to Aroeira et al., the brightness of the thawed meat should be 45.15 ± 2.49. The effect of the coating on the color of the meat can be ruled out, as the brightness is lower for thawed meat, whether it has been protected by the coating or not. Oxidative color changes of myoglobin played a role rather than a gelatin coating. In the same study, the average value of a* thawed meat is reported to be in the interval of 15–21. In this experiment, the value of a* fell within this interval, and only in some cases was it higher than the indicated variance. However, higher values occurred for both coated and uncoated samples. The authors also state that the optimal value b* of thawed meat is 15. In this experiment, the b* value was higher in all thawed samples. The effect of the coating was not observed in this case either, because the values of the yellow-blue shades were similar to the samples with and without the coating. [32]. Sales et al. found that the values obtained before and after thawing differed only minimally, with a decrease, which is the opposite trend from the one in our study [27]. Cheng et al. measured the brightness of the sample before freezing at approximately 50; after thawing, the brightness dropped to 45. The redness value decreased slightly, while b* showing the position on the axis between blue and yellow increased its value deeper into the yellow spectrum [25]. Kim et al. observed, among other properties, a change in color on the surface of the meat. From the obtained values, they did not find any significant color changes due to freezing and thawing of the meat [29].

In the case of Table 7 and Table 8, the results are according to our expectations; no negative effect on the color of the meat was observed on the samples. The color corresponds to the conditions and duration of meat storage and is thus suitable for consumption.

### 3.5. Weight Loss

The values obtained are given in Table 9 and are divided according to the type of coating used, the freezing temperature and the thawing method.

**Table 9 polymers-14-03094-t009:** Weight loss (%) of meat after thawing with and without coating depending on the method of freezing and thawing.

Freezing/Thawing	Gelatin/Glycerol (%)				
	3/10	3/20	5/10	5/20	0/0
−18 °C/R	0.65	0.54	0.81	0.88	0.31
−80 °C/R	0.26	0.39	0.33	0.49	0.24
−18 °C/MW	2.20	2.65	3.37	2.61	1.51
−80 °C/MW	1.11	1.53	1.32	4.18	0.67

R—refrigerator, MW—microwave oven.

The greatest loss of meat frozen at −18 °C and slowly thawed in the refrigerator was observed in a sample with 5% gelatin and 20% glycerol. On the contrary, the lowest loss occurred in the blank sample, on which no protective coating was applied. Only slight weight losses can be expected with fast freezing and slow thawing. In this part of the experiment, the greatest weight loss was observed in the sample with a coating of 3% gelatin and 10% glycerol. The lowest weight loss occurred again in the sample on which no protective coating was applied. Larger-sized ice crystals formed during slow freezing. If the meat is thawed quickly, in this case large ice crystals melt rapidly, which disrupts the structure of the meat fiber and leads to a large weight loss of the meat. Overall, the weight loss during any rapid thawing will always be greater compared to slow thawing. Therefore, the weight losses in this set of samples are higher than in the previous samples. Freezing at a lower temperature did not result in such large losses as slow freezing. The largest decrease was measured in a sample with 5% gelatin and 20% glycerol. Slight weight loss occurred in the uncoated sample.

Table 10 shows the weight losses of the samples with crosslinked coating and uncoated samples (0/0/0) and is divided according to the freezing temperature and the thawing method.

**Table 10 polymers-14-03094-t010:** Weight loss of meat (%) after thawing with and without crosslinked coating depending on the method of freezing and thawing.

Freezing/Thawing		Gelatin/Glycerol/Glutaraldehyde (%)				
	8/10/1				Average	0/0/0
−18 °C/R	1.57	1.47	0.80	0.42	1.07	3.02
−80 °C/R	0.25	0.26	0.38	0.44	0.33	0.42
−18 °C/MW	1.33	2.17	0.84	2.31	1.66	3.84
−80 °C/MW	0.86	0.54	0.67	0.46	0.63	1.75

R—refrigerator, MW—microwave.

During rapid defrosting in the microwave, bigger weight losses occurred again, compared to slow thawing in the refrigerator at 4.0 ± 2.0 °C. It can be assumed that the muscle fibers were broken by the sharp edges of the fast-melting ice crystals; this is especially the case for samples frozen at −18 °C, the dimensions of which were many times larger than the crystals formed at −80 °C. Such a process of melting ice crystals caused higher losses in samples frozen at −18 °C than at −80 °C.

Slow freezing is not a suitable method due to the formation of large ice crystals. However, if the meat is frozen in this way, it should be slowly thawed in the refrigerator. Large weight losses occur when such frozen meat is rapidly thawed in a microwave oven by the action of high temperature. For meat frozen at −80 °C, the best method is thawing in the refrigerator. The second most suitable way of storing meat is again at −80 °C and thawing in the microwave, followed by freezing at −18 °C and thawing in the refrigerator.

Figure 1 for comparison shows the average weight losses of coated and uncoated samples depending on the method of freezing and thawing; in addition to the combination of lower temperature and slow thawing in the refrigerator, lower weight losses were achieved in the gelatin-coated samples. It can also be seen from Figure 1 that the freezing temperature has the greatest effect on the losses. By comparing the weight losses of the gelatin-coated and uncoated samples, it can be assumed that the protective coating affected the permeability of the meat juice and to some extent protected the meat from weight loss. Although the values obtained are larger than in the first part of the experiment, such weight losses of thawed meat are still acceptable in comparison with scientific articles.

Rahman et al. claims that acceptable weight losses should not exceed 3.49% [33]. Oliveira et al. claims that 3.30% is the limit of acceptable meat weight loss [34]. Leygonie et al. in their study reported weight loss of up to 5.5% [9]. Although it was ostrich meat, according to several experts, this meat is similar to beef. These values were determined for meat thawed in the refrigerator. Oliveira et al. set the weight loss for meat thawed in the microwave to 7.29% [34]. The weight losses measured in this experiment are lower. Freezing and thawing temperatures, as well as a protective gelatin coating, definitely had an effect on small weight losses.

Şahin et al., in their study, observed, among other things, weight loss after thawing in two groups. In meat from animals heavier than 504 kg, the decrease was 4.35% greater than in the group up to 503 kg. Loss values were generally higher than in this study due to differences in methodology and meat used [35]. In 2020, Sales et al. found that beef meat in this experiment had a loss of about 7% after thawing, which is 3% higher than in this study [27]. In a study by Cheng, A. et al., the weight loss after the first cycle was about 4%, which corresponds to the results of the uncoated samples of this research [25].

## 4. Conclusions

The study contributes to the expansion and possible development of innovative and promising methods of preserving meat with an edible polymer coating based on gelatin prepared from chicken collagen using the method of enzymatic extraction of gelatin, which was also developed by the authors of this paper. The weight loss of beef—beef sirloin steaks—after freezing and then thawing the meat was studied. Both basic and advanced methods for monitoring meat quality during these preservation methods were employed; in addition to weight loss, physico-chemical properties of meat such as pH, texture (measured by WBSF) and color of beef steaks were also observed. Coated meat samples with the addition of plasticizer and crosslinker showed lower weight loss after thawing than uncoated samples. At the same time, no negative effect was found on other technological parameters of the meat, such as pH, color or texture. The hypothesis about the positive contribution of a coating prepared from chicken paws was confirmed. Chicken paws represent a nontraditional and, up to this day, little-used type of gelatin, which have not yet been processed to a greater extent on a global scale. This can be considered uneconomical and unfriendly to the environment given the large-scale production of poultry meat. The application of this type of coating has several advantages. Above all, this represents a completely new application potential of chicken gelatin prepared by an environmentally friendly procedure. A gelatin coating prepared from unused chicken byproducts improves the quality of the meat due to the retention of water and nutrients in the meat using the preservation method of freezing and subsequent thawing before gastronomic processing into a quality dish—in this case, beef steaks, where it is necessary to preserve juiciness of the meat and supply of nutrients for the consumer. Higher freezing temperatures—deep or shock freezing (which are beneficial for preserving meat)—reduce weight loss and thus help us in the economy of food preparation with a subsequent possible effect on saving the environment while maintaining a gentle approach and not wasting food. Moreover, the reduction in the weight loss of the meat after thawing will have a positive effect on its quality; by preserving the water and nutrients in the meat, juiciness is also preserved. Without a protective coating, after defrosting the meat, a larger amount of water (meat juice) leaves the meat together with other substances present, such as proteins, minerals, vitamins, sensory active substances, etc. In further research, meat packaging from other alternative sources of gelatin and other cross-linking agents will be studied, which will further help to increase the quality of meat while simultaneously ensuring a circular economy with the possible use of hitherto unused animal protein sources.

## Figures and Tables

**Figure 1 polymers-14-03094-f001:**
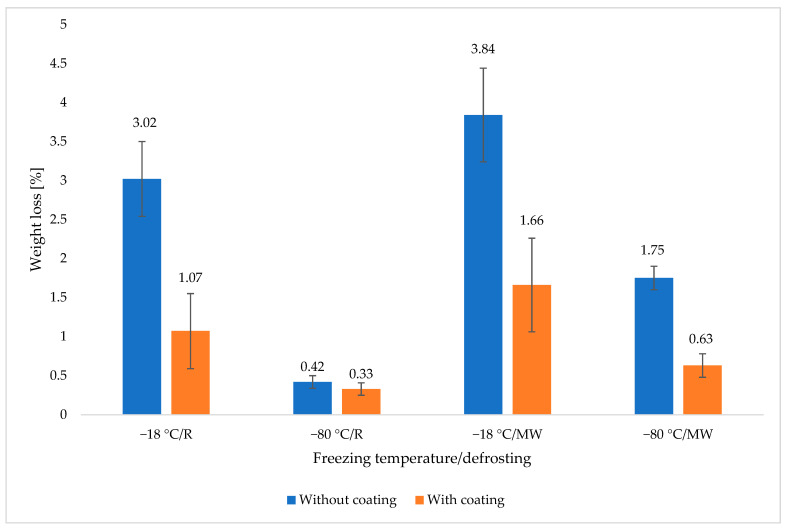
Weight loss of meat without coating and with crosslinked coating depending on the method of freezing and thawing.

## Data Availability

The data presented in this study are available on request from the corresponding author.

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
