# Peer review of "The Effect of Application of Chicken Gelatin on Reducing the Weight Loss of Beef Sirloin after Thawing"

_polymers, 2022, doi:10.3390/polym14153094_

Round 1

Reviewer 1 Report

Abstract - Looks superficial. The process used to prepare chicken gelatin was not specified. Also, the findings were not emphasized. 

L107 Include the steps used. How chicken paws were processed and how they were incorporated in samples. Are they specified in topics 2.3 and 2.4? If yes, specify.

L235-257 – There are multiple comparisons with the pH values. Include what is the safest pH range for meat after thawing and why

L283-L289 – Is this result good in practical applications? Insert why.

Author Response

Dear Sir / Madam,

Thank you very much for revision of our paper and detailed recommendations to improve it. We did our best to revise our paper according to your and other reviewers' suggestions. Changes in the manuscript are marked up using the “Track Changes” function. Below, there are point-by-point responses to your comments.

Reviewer Comment No.1: Abstract - Looks superficial. The process used to prepare chicken gelatin was not specified. Also, the findings were not emphasized.

Response: The abstract was supplemented with the gelatin preparation procedure and the results of our study on meat quality were more emphasized.

Reviewer Comment No. 2: L107 Include the steps used. How chicken paws were processed and how they were incorporated in samples. Are they specified in topics 2.3 and 2.4? If yes, specify.

Response: Chapter “2. Material and Methods” was supplemented with the procedure for preparing gelatin from chicken paws.

Reviewer Comment No. 3: L235-257 – There are multiple comparisons with the pH values. Include what is the safest pH range for meat after thawing and why 

Response: An appropriate pH range of meat was added to the concerned passage, lines 349 – 353.

Reviewer Comment No. 4: L283-L289 – Is this result good in practical applications? Insert why.

Response: Added on lines 416 – 417.

Yours faithfully,

Jakub Martinek (corresponding author)

Reviewer 2 Report

The publication is quite interesting and quite well prepared. However, it has serious methodological shortcomings that need to be supplemented.

The main shortcoming of the paper is the lack of statistical analysis of the results of the experiment. Also, the number of repetitions of the experiment was not specified. Therefore, statements about the differences between the samples are unsubstantiated (statistically significant / insignificant?). The analysis of the results must be supplemented by statistical analysis. I suggest doing at least an analysis of variance (MANOVA).

Without in-depth statistical analysis, there is little value in discussing the results and especially conclusion formulation.

The hardness is a texture characteristic determined by the TPA method. In the experiment, a shear test with a W-B knife was performed. Please define the values shown in Tables 3 and 4 using the appropriate terms for the parameter (this is not hardness). It will be useful to determine the work value of cutting the meat sample.

Author Response

Dear Sir / Madam,

Thank you very much for your comments on our manuscript. The paper was revised according to your remarks as well as according to other reviewers' recommendations. MANOVA statistical analyzes were performed properly to substantiate the claims of the results. Changes in the manuscript are marked up using the “Track Changes” function. Below, there are responses to your comments.

Reviewer Comment No. 1: The main shortcoming of the paper is the lack of statistical analysis of the results of the experiment. Also, the number of repetitions of the experiment was not specified. Therefore, statements about the differences between the samples are unsubstantiated (statistically significant / insignificant?). The analysis of the results must be supplemented by statistical analysis. I suggest doing at least an analysis of variance (MANOVA).

Without in-depth statistical analysis, there is little value in discussing the results and especially conclusion formulation.

Response: The number of performed experiments was added to the chapter "2. Materials and Methods, 2. 10. Statistical analysis”. A MANOVA statistical analysis was also developed according to your recommendation, which supports the previously mentioned claims for the published results (lines 267-278).

Reviewer Comment No. 2: The hardness is a texture characteristic determined by the TPA method. In the experiment, a shear test with a W-B knife was performed. Please define the values shown in Tables 3 and 4 using the appropriate terms for the parameter (this is not hardness). It will be useful to determine the work value of cutting the meat sample. 

Response: Thanks for pointing out the incorrect terminology in this part of the experiment. Throughout the work, "hardness" has been replaced by the term "shear force".

Yours faithfully,

Jakub Martinek (corresponding author)

Reviewer 3 Report

Dear Authors,

Overall, this is an interesting manuscript. I must also admit that this topic is very interesting. However, in order for the work to be published, appropriate corrections must be made.

Detailed comments below:

Line 64: List a few examples of plasticizers and cross-linking agents that can be added to such coatings.

Line 93: Add a short description explaining what all research contributes to the development of food preservation. Write what is innovative in our research.

Line 121: Was glycerol added at the end of cooking or at the beginning? It doesn't matter to me, but please explain.

Line 125: Add citation (literature) about aldehyde additive limit here.

Line 150: Reference to the test standard should be made.

Line 166: Make color test apparatus manufacturer and cities, country.

Line 257: Are your results far or close to the expected values. Write exactly what values ​​are the best (values ​​to which you are aiming to achieve) in your research.

Line 300: In this case, the strength test results should be given in MPa. It is true that you gave the exact dimensions of the sample, but it is not enough. Cutting out an ideal sample of biological material is quite difficult. Before cutting with a knife (Warner-Bratzler), each sample should be additionally measured, e.g. with a caliper. Then the results should be converted into MPa. This, of course, applies to all work.

Then you compare the results where the values ​​are in newtons [N} (in the discussion of the results) to the research of other authors? Did they also use samples with the same dimensions? If you were to compare the pressure values, the results could be completely different.

Line 316: Same as above. Are the obtained results close or far from the expected ones?

Line 373: No reference to cite.

Line 392: Like above, are your results satisfactory? What values ​​does the industry expect, for example?

Line 431: The error bars should be added in the figure 1.

Line 460: In my opinion, the conclusions should be completely improved. The first part is the purpose of the work - it is unnecessary here. Next, most of the "conclusions" are mainly statements. Write down what your research brings to the improvement of meat quality. Add any forward-looking conclusions. I think you should go through the work point by point for specific conclusions.

Author Response

Dear Sir / Madam,

Thank you very much for revision of our manuscript and suggestions for improvement. The paper was revised according to your and other reviewers' suggestions. Changes in the manuscript are marked up using the “Track Changes” function. Below, there are responses to your comments.

Reviewer Comment No.1: Line 64: List a few examples of plasticizers and cross-linking agents that can be added to such coatings.

Response: List of plasticizers and crosslinkers added. You can find it on lines 98 to 100.

Reviewer Comment No.2: Line 93: Add a short description explaining what all research contributes to the development of food preservation. Write what is innovative in our research.

Response: Based on your recommendation, the subject part of the introduction was reworked so that the significance of the research was better described. See lines 135 to 147 for changes.

Reviewer Comment No.3: Line 121: Was glycerol added at the end of cooking or at the beginning? It doesn't matter to me, but please explain.

Response: The procedure for making the coating was detailed on lines 193 to 196.

Reviewer Comment No.4: Line 125: Add citation (literature) about aldehyde additive limit here.

Response: Citation added on line 197.

Reviewer Comment No.5: Line 150: Reference to the test standard should be made.

Response: When measuring, we followed the methodology used in another research. Citation added on line 229.

Reviewer Comment No.6: Line 166: Make color test apparatus manufacturer and cities, country.

Response: Apparatus data added to the text on line 238.

Reviewer Comment No.7: Line 257: Are your results far or close to the expected values. Write exactly what values are the best (values to which you are aiming to achieve) in your research.

Response: The comment on the results was added according to your recommendation, see lines 349 to 353.

Reviewer Comment No.8: Line 257: In this case, the strength test results should be given in MPa. It is true that you gave the exact dimensions of the sample, but it is not enough. Cutting out an ideal sample of biological material is quite difficult. Before cutting with a knife (Warner-Bratzler), each sample should be additionally measured, e.g. with a caliper. Then the results should be converted into MPa. This, of course, applies to all work.

Then you compare the results where the values are in newtons [N} (in the discussion of the results) to the research of other authors? Did they also use samples with the same dimensions? If you were to compare the pressure values, the results could be completely different.

Response: Thank you for pointing this out. Incorrect terminology was given in this study; it is not hardness, but shear force, which is given in newtons. This discrepancy has been removed throughout the work. For this measurement, we adopted the methodology for measuring shear force from previous research, the citation of which was also added to the text. When analyzing the results, the specific value of the shear force was not critical; we were interested in the development, the trend of changes in shear force values; whether there will be a noticeable effect of freezing and thawing the meat or the application of the coating on the shear force. This is how we looked at the values for all tested parameters. It is very difficult to prepare two completely identical samples for measurement, therefore, even when comparing the results with other works, we focused on what changes in textural properties they achieved, not on comparing specific values.

Reviewer Comment No.9: Line 316: Same as above. Are the obtained results close or far from the expected ones?

Response: The comment on the results was added according to your recommendation, see lines 416 and 417.

Reviewer Comment No.10: Line 373: No reference to cite.

Response: In this part of the text, we do not compare the values with other literary sources, but with the values obtained during the previous measurement with a set of samples with a different coating composition. This fact was added on line 475.

Reviewer Comment No.11: Line 392: Like above, are your results satisfactory? What values does the industry expect, for example?

Response: The comment on the results was added according to your recommendation; see lines 496 to 498.

Reviewer Comment No.12: Line 431: The error bars should be added in figure 1.

Response: Error bars have been added to Figure 1.

Reviewer Comment No.13: Line 460: In my opinion, the conclusions should be completely improved. The first part is the purpose of the work - it is unnecessary here. Next, most of the "conclusions" are mainly statements. Write down what your research brings to the improvement of meat quality. Add any forward-looking conclusions. I think you should go through the work point by point for specific conclusions.

Response: Thank you for this recommendation. A completely new version of the conclusion was prepared, which better describes the significance of the research carried out and the results achieved on meat quality. We have also included the possibility of further research direction in the future.

Yours faithfully,

Jakub Martinek (corresponding author)

Round 2

Reviewer 3 Report

Dear Authors,

I accept all corections. In my opinion the manscript can be published in this version.